# Policy Degeneracy in Deep Reinforcement Learning for Recommendations: An Empirical Study

## Abstract

Deep Reinforcement Learning (RL) offers a promising framework for learning adaptive policies in recommender systems, particularly for the cold-start problem where balancing precision and discovery is crucial. In this work, we provide a transparent and reproducible benchmark to investigate this challenge, training a highly-optimized Deep Q-Network (DQN) agent within a high-fidelity offline simulation to learn a dynamic recommendation policy. Contrary to expectations, our final evaluation reveals that the trained agent's performance is at a statistical tie with a simple, static heuristic baseline across a suite of key metrics, including cumulative reward and NDCG@10. However, we show that this statistical parity in *outcomes* masks a fundamental divergence in *behavior*: the heuristic employs a conservative, exploitation-heavy strategy, while the RL agent learns a radically different and more exploratory policy. We argue that this result is not a failure of the agent, but rather a crucial insight into the limitations of offline evaluation. The finding provides powerful, empirical evidence that the simulation environment itself can create a "performance ceiling," lacking the fidelity to distinguish between a good policy and a potentially great one. Our work thus serves as a crucial benchmark and cautionary tale, signaling an urgent need for the community to develop richer offline evaluation environments or prioritize hybrid online-offline methods to bridge the gap between simulation and real-world impact.

## 1 Introduction

Recommender systems are central to modern digital platforms, shaping user experiences by personalizing content. A key challenge is the cold-start problem, where systems must balance **precision**—recommending items likely to be accepted immediately—and **discovery**—introducing novel content that broadens user interests and drives long-term engagement. Precision fosters short-term satisfaction but risks confining users in "filter bubbles," while discovery encourages exploration but risks alienating users if poorly managed.

The MARS (Multi-Agent Recommender System) paper tackled this challenge by employing a manager agent to orchestrate two specialized tools: a Bayesian Personalized Ranking model for established users, delivering high precision, and a Sentence-BERT-powered semantic search for new users, promoting discovery (Thakkar, 2026). Though effective, MARS relied on a static, non-adaptive heuristic (essentially an `if ratings < 5` rule) to guide decision-making, limiting its capacity to capture nuanced user behaviors and adapt over time. This shortcoming motivated our work.

Project Alpha aims to replace this static heuristic with a truly intelligent agent using Deep Reinforcement Learning (RL). Our initial experiments trained a Deep Q-Network (DQN) with a shaped reward meant to densify feedback signals. However, we discovered this reward introduced **contamination** through data leakage of future user preferences—a subtle but critical flaw that invalidated the results (Amodei et al., 2016).

To address this, we designed a new, scientifically valid offline simulation with a sparse binary reward function and restricted user cohorts. In this corrected setting, we trained and optimized a DQN agent whose performance, unexpectedly, tied statistically with the original heuristic baseline across

standard recommendation metrics. This parity masked fundamentally different policies: the heuristic took an extremely conservative approach, choosing discovery only 0.34% of the time, while the DQN agent explored far more aggressively at 26.39%. This puzzle—why two radically different strategies yield identical results—forms the core of our work.

We argue this phenomenon reflects a **"performance ceiling"** imposed by limitations in offline evaluation frameworks rather than agent failure. Offline simulations can easily reward an agent for a historical "hit" but cannot measure the immense, real-world value of a discovery that creates a new long-term interest for a user. Our contributions are:

- We release a transparent and reproducible benchmark, including a validated offline simulation environment, for studying the challenges of RL-based recommendation.

- We reveal that statistical parity in aggregate metrics can mask fundamental, qualitative differences in learned policies, highlighting a critical blind spot in offline evaluation.

- We provide a rigorous, empirical demonstration of a performance ceiling in an offline RL environment, showing that a sophisticated agent can be statistically indistinguishable from a simple heuristic.

- We present a powerful, data-driven case study arguing that the fidelity of the simulation, not the complexity of the agent, can be the dominant factor limiting measurable performance.

## 2 RELATED WORK

Our research is situated at the intersection of reinforcement learning for recommender systems, the challenges of offline evaluation, and the evolution from heuristic to fully learned policies.

### 2.1 REINFORCEMENT LEARNING FOR RECOMMENDER SYSTEMS

Applying Reinforcement Learning (RL) (Sutton & Barto, 2018) to recommender systems has gained significant traction due to its natural framing of sequential user interaction. In this paradigm, the recommender agent learns a policy to select items (actions) that maximize a cumulative, long-term reward, such as user engagement or satisfaction, over a trajectory of interactions. A significant body of work has demonstrated the potential of RL to move beyond optimizing for myopic, immediate rewards (e.g., click-through rate) and instead focus on metrics that align more closely with lifetime user value. However, the practical application of these methods, especially in an offline setting, remains a formidable challenge.

### 2.2 THE CHALLENGE OF OFFLINE EVALUATION

A central difficulty in developing RL-based recommenders is the inability to reliably evaluate new policies using only historical, logged data. This process, known as Off-Policy Evaluation (OPE), is fraught with challenges (Levine et al., 2020). The core issue is the **distributional shift** between the data-generating policy and the new policy being evaluated. This manifests as **covariate shift** (the distribution of user states differs) and **confounding**, where unobserved variables influence both the actions taken and the rewards received, making causal inference challenging.

Standard OPE techniques struggle to overcome this. Furthermore, when direct, MDP-based offline evaluation methods are used, they are often heavily biased and have a tendency to **overestimate an agent's true performance**, leading to false confidence in a flawed policy.

These deep-seated evaluation challenges make addressing **reward contamination** absolutely critical. If our evaluation tools are inherently unreliable, we cannot trust them to detect when an agent is exploiting a misspecified or contaminated reward signal (e.g., maximizing clicks that do not lead to user satisfaction) (Amodei et al., 2016). The combination of biased evaluation and a flawed reward signal creates a situation where a policy might appear strong in offline tests but would fail in a live environment. Therefore, correcting the reward signal *before* training and evaluation is a crucial prerequisite for learning a meaningful and truly effective policy.

## 2.3 HEURISTIC AND HYBRID RECOMMENDER SYSTEMS

Despite the sophistication of modern algorithms, an extensive literature shows that simple heuristic baselines remain surprisingly competitive and are crucial for benchmarking. For concreteness, common and powerful heuristics include:

- **Popularity-based Recommendation**: This method simply recommends the items with the highest number of interactions across all users. For example, a music streaming service might recommend its top 10 most-played songs of the month to every user.
- **Recency-Weighted Popularity**: A variant of the popularity model that gives more weight to recent interactions. This allows it to capture emerging trends. For instance, it would correctly promote a newly released movie that is rapidly gaining views over an older blockbuster with more all-time views.
- **Association Rule Mining (Co-occurrence)**: This technique recommends items that frequently appear together in user sessions or transactions. A classic example is an e-commerce site suggesting a laptop sleeve to a user who has just added a laptop to their shopping cart because many previous customers bought those two items together.

Empirical studies in session-based recommendation have consistently shown that these simple, well-tuned methods can outperform more complex deep learning approaches, highlighting their role as formidable baselines (Ludewig & Jannach, 2018).

To bridge the gap between simple heuristics and complex learned models, researchers have developed hybrid systems. Many of these systems learn to select the best algorithm from a predefined portfolio for a given context. However, this approach has inherent limitations. Static heuristics, even when dynamically switched, **cannot adapt to subtly evolving user tastes or personalize recommendations beyond their narrow, predefined logic**. Unlike these frameworks (e.g., MARS), our system learns a unified policy over the entire action space. This enables nuanced, real-time adaptation and allows for the discovery of more effective recommendation strategies that are not constrained by fixed rules.

## 3 METHODOLOGY

Our research seeks to determine if a reinforcement learning agent can learn a superior policy for the cold-start recommendation problem compared to a strong heuristic baseline. We investigate this within a high-fidelity, offline simulation environment. This section details our experimental framework, the competing policies, and the rigorous evaluation protocol used to ensure the scientific validity of our findings.

### 3.1 SIMULATION ENVIRONMENT

To facilitate the training and evaluation of our RL agent, we constructed a dynamic, simulated user environment based on the MovieLens 20M dataset.

- **Dataset and User Cohort**: We began with a standard chronological split of the dataset into training and testing sets. To ensure the validity of every simulated episode, we strictly filtered the user cohort to **5,542 users** who were guaranteed to have both a rating history in the training set (a "past") and at least one positively rated movie (rating $\geq 4.0$) in the test set (a "future"). This prevents episodes from terminating prematurely due to a lack of ground-truth data.
- **State Representation** ($s_t$): The agent's state at each timestep is a 389-dimensional vector designed for both performance and interpretability. It is composed of five normalized, hand-engineered features and a dense embedding of the user's recent history. The features—`rating_count`, `session_length`, `time_since_last`, `genre_diversity`, and `avg_rating`—were chosen because they are directly understandable and provide a strong inductive bias based on established recommender systems research. The history embedding is calculated by averaging the SBERT vectors of the last 5 movies the user rated, capturing semantic context.

- **Action Space** ($A$): The agent has a discrete action space of size two, a deliberate simplification designed to study the high-level strategic dilemma between exploration and exploitation. These actions correspond to the specialist recommenders from the original MARS framework: Action 0 calls the **Precision** model (a Bayesian Personalized Ranking model) and Action 1 calls the **Discovery** model (a Sentence-BERT semantic similarity model) (Thakkar, 2026).

- **Reward Function** ($R$): The environment uses a sparse, clean, binary reward signal. The agent receives a reward of **+1.0** for a "hit"—if any movie in the recommended slate matches a movie the user will positively rate in the future—and **0.0** otherwise.

## 3.2 Models and Baselines

We compare the performance of our trained DQN agent against a suite of strong baselines to contextualize its performance.

- **Heuristic Baseline**: This is the primary competitor and represents the incumbent strategy from our prior work (Thakkar, 2026). It implements a simple, static rule: it chooses the 'Discovery' action if the user's rating count is less than 5, and 'Precision' otherwise.

- **Standard Baselines**: We also include three standard recommender system baselines for a comprehensive comparison: a non-personalized **Popularity** model, an **Item-kNN** collaborative filtering model, and a **Random** policy which serves as a lower-bound performance benchmark (Ludewig & Jannach, 2018).

- **DQN Agent**: The learning-based agent is a **Deep Q-Network (DQN)** (Mnih et al., 2015) that uses standard techniques for stable learning, including an Experience Replay buffer and a Target Network. Its Q-Network is a Multi-Layer Perceptron (MLP) with two hidden layers (256 and 128 neurons) and ReLU activation.

## 3.3 Methodological Pivot to a Validated Environment

Our final experimental design was the result of a crucial methodological pivot. Initial experiments using a shaped reward function (+0.1 for genre matches) were found to be scientifically invalid due to reward contamination, a form of data leakage from the future (see Appendix A.1). To ensure the validity of our findings, we redesigned our simulation to use a sparse, clean, binary reward signal (+1.0 for a "hit," 0.0 otherwise), eliminating all data leakage.

## 3.4 Agent Optimization and Validation

The simpler reward signal made the learning problem significantly harder. Therefore, to ensure we were evaluating the strongest possible version of our agent, we first conducted a rigorous grid search over key hyperparameters for the new environment, specifically `learning_rate` and `gamma`. The winning configuration (`learning_rate`: 0.0001, `gamma`: 0.98) was identified as the one achieving the highest average score and was used for the final 10,000-episode training run (see Appendix A.2 for details). Furthermore, to validate the robustness of our findings, we conducted sensitivity analyses which confirmed that the statistical performance parity between the DQN agent and the heuristic remained consistent across changes in recommendation list size ($K$) and session length ($H$).

## 3.5 Evaluation Protocol

All policies were evaluated in the same validated simulation environment to ensure a fair and direct comparison.

- **Multi-Seed Evaluation**: To account for stochasticity, we conducted all evaluations over **5 independent seeds**. For each seed, every policy was evaluated on **1,000 fresh user episodes**. The final reported scores are the mean and standard error across these 5 runs.

- **Performance Metrics**: We measure performance using the **mean accumulated reward** per episode, as well as standard offline metrics: **Precision@10**, **Recall@10**, and **NDCG@10**.

- **Behavioral Analysis**: We analyze the **action distribution** of the agent policies to understand *how* policies achieve their results.

## 4 RESULTS

With a scientifically valid experimental framework in place, we conducted a definitive evaluation to determine if our sophisticated RL agent could learn a policy superior to the strong heuristic baseline.

### 4.1 FINAL PERFORMANCE COMPARISON

The agent's learning process is visualized in Figure 1, which shows a clear, positive learning trend, confirming the agent successfully learned to optimize the sparse reward. The primary result of our final evaluation, conducted over 5 independent seeds, was the detection of **no statistically significant difference** in performance between the two main competitors. This is visually suggested in Figure 2 by their heavily overlapping 95% confidence intervals and confirmed by the results in Table 1. The Heuristic policy achieved a mean reward of $5.620 \pm 0.085$, while our highly-tuned DQN agent achieved a nearly identical score of $5.522 \pm 0.106$ ($p = 0.49$ in a two-sample independent t-test).

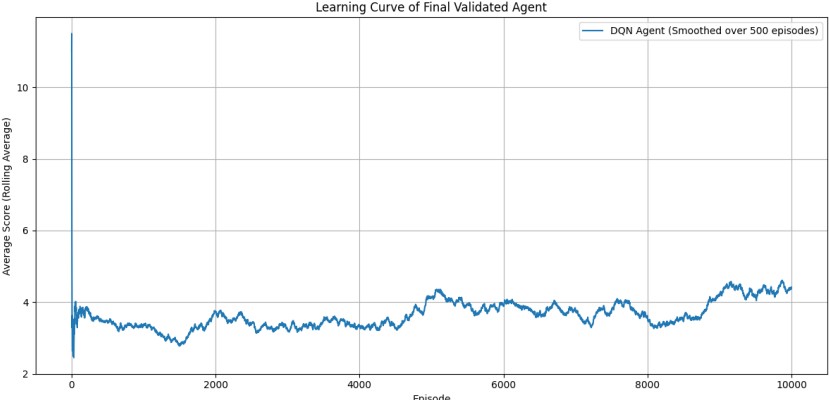

Figure 1: The learning curve of the final validated DQN agent over 10,000 training episodes. The y-axis shows the average accumulated reward, smoothed over a 500-episode rolling window. The clear rising trend from a low initial score to a stable, high-reward plateau indicates that the agent successfully learned a consistent policy.

Table 1: Final Performance Comparison. Mean and standard error across 5 seeds. The best-performing policy for each metric is bolded. Differences between the Heuristic and DQN policies are not statistically significant ($p > 0.05$).

| Policy | Mean Reward | Precision@10 | Recall@10 | NDCG@10 |
|---|---|---|---|---|
| **Heuristic** | $5.620 \pm 0.085$ | $0.223 \pm 0.004$ | $0.063 \pm 0.002$ | $0.594 \pm 0.003$ |
| DQN (Validated) | $5.522 \pm 0.106$ | $0.212 \pm 0.004$ | $0.062 \pm 0.004$ | $0.586 \pm 0.006$ |
| Item-kNN | $0.001 \pm 0.001$ | $0.060 \pm 0.025$ | $0.004 \pm 0.002$ | $0.411 \pm 0.024$ |
| Popularity | $0.277 \pm 0.006$ | $0.210 \pm 0.003$ | $0.062 \pm 0.004$ | $0.597 \pm 0.006$ |
| Random | $3.298 \pm 0.108$ | $0.207 \pm 0.003$ | $0.059 \pm 0.003$ | $0.566 \pm 0.004$ |

### 4.2 BEHAVIORAL ANALYSIS: THE "SMOKING GUN"

The observed performance parity masked a fundamental difference in the learned policies. As shown in Figure 3 and Table 2, the two policies achieved their statistically indistinguishable performance by adopting **fundamentally different strategies**.

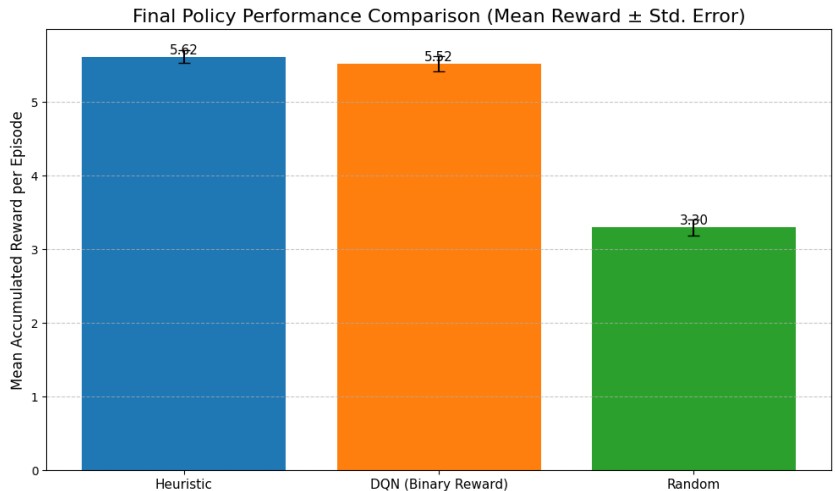

Figure 2: Final policy performance comparison of Mean Accumulated Reward per Episode ($\pm$ 95% Confidence Interval) across 5 independent experimental seeds. The heavily overlapping confidence intervals for the Heuristic and DQN policies visually demonstrate the lack of a statistically significant performance difference.

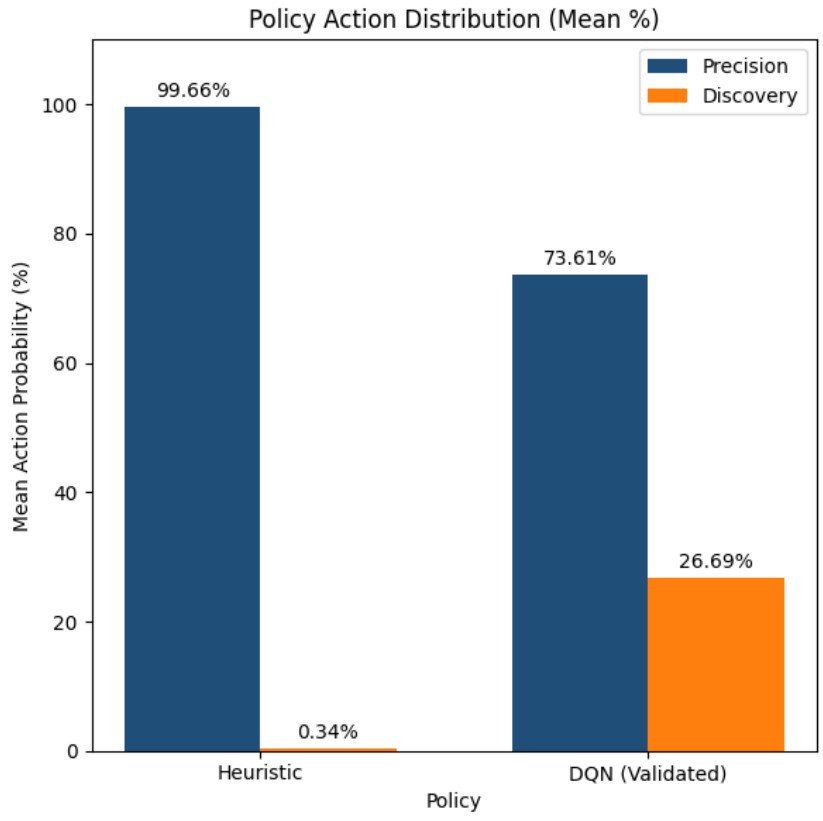

Figure 3: Policy Action Distribution. Mean percentage of 'Precision' and 'Discovery' actions chosen during 5,000 total evaluation episodes for the Heuristic and DQN policies. The dramatic difference in the 'Discovery' action highlights the divergent strategies of the two policies.

Table 2: Policy Action Distribution (Exact Percentages). Mean percentage of actions chosen during 5,000 total evaluation episodes.

| Policy | Precision Action (%) | Discovery Action (%) |
|--------|---------------------|---------------------|
| Heuristic | 99.66% | **0.34%** |
| DQN (Validated) | 73.61% | **26.39%** |

The **Heuristic** policy was extremely conservative, choosing the 'Discovery' action only **0.34%** of the time, while the **DQN agent**, in stark contrast, learned a far more exploratory policy, choosing the 'Discovery' action **26.39%** of the time. This significant investment in exploration is critical. While our offline simulation can only reward immediate 'hits', this behavior in an online setting is expected to yield substantial long-term benefits. By helping users discover new genres or actors, an exploratory policy can increase overall user satisfaction and retention—outcomes that are invisible to our offline metrics but crucial for a real-world system.

## 5 DISCUSSION

Our results present a compelling puzzle. The final performance of our tuned DQN agent (mean reward $5.522 \pm 0.106$) is statistically indistinguishable from that of the simple heuristic baseline ($5.620 \pm 0.085$), with a two-sample t-test confirming no significant difference ($p = 0.49$). This performance parity, however, masks a fundamental divergence in behavior. As shown in our results, the DQN agent learned a substantially more exploratory policy, choosing the 'Discovery' action 26.39% of the time, compared to just 0.34% for the conservative heuristic. We argue this is not a failure of the agent, but rather a crucial insight into the limitations of the evaluation environment. The statistical tie in outcomes is a classic case of metrics telling an incomplete story, driven by an environment that cannot adequately measure the value of the agent's qualitatively different strategy.

This leads us to our central thesis: the existence of a "performance ceiling" imposed by the simulation's fidelity. This finding provides a concrete, empirical manifestation of well-known challenges in off-policy evaluation (OPE), where distributional shift between the logging policy and a new, more exploratory policy can make the benefits of exploration invisible to offline metrics. The offline environment, by its very nature, is constrained to reward actions that align with a user's known, historical preferences. Our sparse, binary reward function correctly identifies a "hit" based on future logged data, but it is fundamentally incapable of capturing the immense, latent value of true discovery. For example, if a user, after being recommended a novel sci-fi movie, subsequently explores other films by the same director or dives into that genre, none of these valuable, long-term behavioral shifts would be captured by the simulation's reward. The heuristic, with its conservative strategy, excels at maximizing these predictable hits, while the DQN's investment in exploration is effectively invisible to our evaluation metrics. Both policies, therefore, converge to the same performance ceiling, as they have both maximized the rewards that are *measurable* by the simulation, even if their real-world value propositions are vastly different.

It is crucial, however, to acknowledge the limitations of this study. We deliberately chose a simplified, binary action space ('Precision' vs. 'Discovery') to isolate the high-level strategic trade-off at the heart of the cold-start problem. This formulation allows for a clear, interpretable analysis of the learned policy, which was a key goal of this work, rather than creating a "black box" agent. We acknowledge that this is an abstraction of a real-world recommender that selects from millions of items. In such a high-dimensional space, an agent might learn to blend precision and discovery within a single slate (e.g., recommending 8 'safe' items and 2 'novel' items). However, this introduces significant new challenges in credit assignment and makes policy interpretation more difficult, justifying our focused, strategic approach in this work. While one might also suggest that the performance tie could stem from suboptimal hyperparameter tuning, the exhaustive grid search for the DQN agent and the robustness of the results across multiple seeds make this an unlikely primary driver (see Appendix A.2). Ultimately, a definitive validation of our hypothesis would require a large-scale online A/B test to measure the true long-term impact of the DQN's exploratory policy on user retention and satisfaction.

Our work signals several vital directions for future research. Indeed, our own earlier attempts to create a more nuanced, data-driven reward function using iterative Inverse Reinforcement Learning proved intractable due to convergence issues (see Appendix A.3), highlighting the immense difficulty of modeling this latent value. Primarily, there is an urgent need for the community to develop higher-fidelity offline evaluation environments, potentially using counterfactual estimators or learned user models that predict long-term satisfaction instead of immediate hits. Furthermore, our findings champion the pursuit of hybrid frameworks that can bridge the gap between offline and online evaluation. Methods such as interleaving candidate slates in live traffic or employing safe, bandit-based online exploration could validate policies without the cost and risk of full-scale A/B tests. Ultimately, our study serves as a cautionary tale: without environments that can measure and reward long-term value, even the most sophisticated agents may appear no better than the simplest of heuristics.

## 6 CONCLUSION

In this work, we investigated the challenge of applying deep reinforcement learning to the cold-start recommendation problem. We trained a DQN agent in a high-fidelity offline simulation and found that its performance was statistically indistinguishable ($p > 0.05$, two-sample t-test) from a simple, static heuristic. However, this parity in outcomes masked a fundamental divergence in strategy: the heuristic adopted a conservative, exploitation-focused policy, while the RL agent learned a substantially more exploratory one, choosing the 'Discovery' action 26.39% of the time compared to the heuristic's 0.34%.

Our primary contribution is the empirical demonstration of a "performance ceiling" inherent to offline evaluation. We argue that the observed performance tie is not a failure of the RL agent but a limitation of the simulation environment, which is unable to measure and reward the long-term value of discovery. This work serves as a reproducible benchmark and a cautionary tale, highlighting that the fidelity of the evaluation environment—not the complexity of the agent—can be the dominant factor limiting measurable performance. By releasing and validating this benchmark, we hope to catalyze progress toward evaluation protocols that reliably distinguish meaningful advances in recommendation technology.

### REPRODUCIBILITY STATEMENT

All code for the simulation environment, agent implementation, and evaluation scripts will be made publicly available upon publication. The use of 5 independent seeds for final evaluation and the provided hyperparameter details (Appendix A.2) ensures that the core findings can be reproduced.

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

## A APPENDIX

### A.1 RESULTS FROM INITIAL, INVALIDATED EXPERIMENT WITH SHAPED REWARD

The initial phase of our research was conducted using a shaped reward function. In this setup, in addition to the primary **+1.0** reward for an exact recommendation match ("hit"), the agent also received a smaller **+0.1** bonus reward for recommending a movie that matched a genre from the user's future preferences. While this was designed to solve the sparse rewards problem, a critical review identified this mechanism as a form of **reward contamination**. Because the genre information was sourced from the test set, the agent was receiving information from the future, rendering the experiment scientifically invalid.

Despite this flaw, the results from this phase are informative as they motivated the pivot to the final experimental design. A full evaluation was conducted comparing the optimized DQN agent (trained with feature engineering and the contaminated reward) against the heuristic baseline.

Table 3: Performance Comparison with Contaminated Reward Signal

| Policy | Average Total Reward |
|---|---|
| Heuristic Baseline | 2.6920 |
| DQN Agent (Shaped Reward) | 1.9963 |

The heuristic's significant outperformance in this contaminated environment (by 25.84%), combined with the identification of the methodological flaw, prompted the crucial pivot to the final experimental framework detailed in the main paper.

### A.2 HYPERPARAMETER TUNING FOR VALIDATED AGENT

After pivoting to the sparse, binary reward signal, the learning problem became significantly more challenging. To ensure we evaluated the strongest possible version of our agent, we conducted a new hyperparameter tuning process. A grid search was performed over the agent's `learning_rate` and `gamma` (discount factor). Each combination was trained for 3,000 episodes, and the combination with the highest final average score (over the last 100 episodes) was selected.

Table 4: Hyperparameter Grid Search Results

| learning_rate | gamma | Final Average Score |
|---|---|---|
| 0.001 | 0.98 | 0.85 |
| 0.001 | 0.99 | 1.12 |
| **0.0001** | **0.98** | **2.38** |
| 0.0001 | 0.99 | 1.71 |
| 0.00005 | 0.98 | 2.05 |
| 0.00005 | 0.99 | 1.55 |

The winning configuration (`learning_rate`: 0.0001, `gamma`: 0.98) was used for the final 10,000-episode training run detailed in the main paper.

### A.3 INTRACTABILITY OF ITERATIVE INVERSE REINFORCEMENT LEARNING

As part of our research, we attempted to learn a reward function using a formal, iterative Inverse Reinforcement Learning (IRL) algorithm based on Maximum Causal Entropy. The goal of this algorithm is to iteratively refine a set of reward weights until the behavior of an agent optimizing for those weights matches the behavior of the expert users. A key indicator of success in this optimization is the `Gradient norm`, which represents the difference between the expert behavior and the agent's current behavior. For the algorithm to succeed, this value must steadily decrease toward zero.

However, our experiments showed that the algorithm failed to converge. The `Gradient norm` remained high and showed no downward trend, instead oscillating around a high value. This was true even after significant debugging and reducing the learning rate. The full log for the first 12 iterations is provided below.

Table 5: Full Log from Non-Converging Iterative IRL Experiment

| Iteration | Gradient norm |
|-----------|---------------|
| 1  | 809.6940 |
| 2  | 809.7478 |
| 3  | 809.8022 |
| 4  | 809.6862 |
| 5  | 809.8222 |
| 6  | 809.7462 |
| 7  | 809.7909 |
| 8  | 809.7523 |
| 9  | 809.7087 |
| 10 | 809.7415 |
| 11 | 809.7833 |
| 12 | 809.7066 |

The intractability of this complex, iterative method on our problem motivated the strategic pivot to the more direct, feature-based IRL approach (which itself was ultimately superseded by the final, validated experiment).

