# OpenReview forum: "POLICY DEGENERACY IN DEEP REINFORCEMENT LEARNING FOR RECOMMENDATIONS: AN EMPIRICAL STUDY"
_ICLR.cc/2026/Conference — Submitted to ICLR 2026_

### Official Review · Reviewer_YovQ · 2025-10-14

**Soundness:** 1
**Presentation:** 1
**Contribution:** 1
**Rating:** 2
**Confidence:** 4

**Summary:**

This manuscript aims to address the precision-exploration trade-off in recommendation cold-start problems by replacing the static heuristic policy of the MARS system with a DQN agent. After correcting data leakage issues in the initial shaped reward design, it conducts offline simulation experiments. Results show the DQN agent exhibits statistically indistinguishable performance from the MARS heuristic but adopts a more exploratory strategy. Notably, the study relies heavily on a MARS-related literature (Thakkar, 2026) labeled as a WSDM'26 conference paper as its core baseline.

**Strengths:**

The experimental design has basic rigor, such as identifying and rectifying data leakage in the shaped reward function. It also provides partial reproducible details, including DQN hyperparameters (e.g., learning rate 0.0001, gamma 0.98) from grid search.

**Weaknesses:**

1. The MARS literature (Thakkar, 2026) cited as the core baseline is labeled WSDM'26, but WSDM'26 remains in review—no formal acceptance/publication proof exists, invalidating its use as "accepted-in-press."
2. Exploration relies on Sentence-BERT-based semantic similarity, limiting it to user-preference-related content. Practical exploration requires covering non-semantically similar new interests, reducing real-world applicability.
3. The study fails to cite KuaiSim[1]. KuaiSim is a foundational benchmark for offline recommendation simulations, and omitting it breaks connections to existing academic context, leaving the study’s simulation design ungrounded in prior advances.

[1] KuaiSim: A comprehensive simulator for recommender systems. NeurIPS'23

**Questions:**

1. Can official proof for the MARS literature (Thakkar, 2026) from WSDM'26 be provided?
2. Why is KuaiSim[1]—a foundational work in offline recommendation simulation—not cited, and how does this omission affect the study’s alignment with prior research?

[1] KuaiSim: A comprehensive simulator for recommender systems. NeurIPS'23

---

### Official Review · Reviewer_tkTF · 2025-10-16

**Soundness:** 2
**Presentation:** 2
**Contribution:** 2
**Rating:** 2
**Confidence:** 4

**Summary:**

This paper investigates deep reinforcement learning (RL) for recommender systems, focusing on whether a DQN agent can outperform a heuristic baseline in an offline cold-start simulation. The authors report that the RL agent achieves statistically indistinguishable performance from a simple heuristic but exhibits qualitatively different behavior (more exploratory vs. conservative).

**Strengths:**

Good writing quality and clear structure.
Transparent reporting of invalid experiments and hyperparameter tuning.
Honest recognition of reproducibility and evaluation challenges in RL-based recommenders.
Inclusion of behavioral policy analysis beyond scalar metrics is thoughtful.

**Weaknesses:**

Demonstrating that RL ties heuristics in a constrained offline simulation is unsurprising and provides limited new understanding. The claimed “performance ceiling” is an expected artifact of restricted data and evaluation bias.

The binary action and reward space are overly reductive and cannot represent realistic recommender dynamics. This simplicity undermines the claimed insight into reinforcement learning behavior.

The experiments rely on one dataset and lack ablations on architecture, reward shaping, or OPE techniques. The absence of comparisons with modern offline RL methods limits credibility.

The proposed “benchmark” lacks diversity and realism, preventing it from being a useful tool for broader community evaluation. The simulation’s simplicity contradicts its framing as a “high-fidelity” environment.

**Questions:**

please refer to weakness part

---

### Official Review · Reviewer_iBgK · 2025-10-30

**Soundness:** 3
**Presentation:** 3
**Contribution:** 3
**Rating:** 8
**Confidence:** 4

**Summary:**

Summary:

In this work, the authors provide a transparent and reproducible benchmark to investigate this challenge, training a highly-optimized Deep Q-Network (DQN) agent within a high-fidelity offline simulation to learn a dynamic recommendation policy. Contrary to expectations,
their final evaluation reveals that the trained agent’s performance is at a statistical tie with a simple, static heuristic baseline across a suite of key metrics, including cumulative reward and NDCG@10. However, they show that this statistical parity in outcomes masks a fundamental divergence in behavior: the heuristic employs a conservative, exploitation-heavy strategy, while the RL agent learns a radically
different and more exploratory policy. They argue that this result is not a failure of the agent, but rather a crucial insight into the limitations of offline evaluation. The finding provides powerful, empirical evidence that the simulation environment itself can create a ”performance ceiling,” lacking the fidelity to distinguish between a good policy and a potentially great one. Their work thus serves as a crucial benchmark and cautionary tale, signaling an urgent need for the community to develop richer offline evaluation environments or prioritize hybrid online-offline methods to bridge the gap between simulation and real-world impact.

Contribution:
* It provides a transparent and reproducible benchmark for studying RL in recommender systems, including a validated offline simulation environment.

* It delivers a powerful, empirical demonstration that statistical parity in competing heuristic and RL models in traditional RS metrics can mask their fundamental, qualitative differences in learned policies.

* It presents a rigorous case study of a "performance ceiling" in offline RL evaluation, arguing convincingly that the simulation's fidelity of the benchmark could potentially hinder the act of distinguishing between a good policy and a potentially great one, thus creating the need of a richer offline evaluation environments construction.

**Strengths:**

1. Methodological Rigor and Transparency: The paper's greatest strength is its honesty. The authors discovered their initial experiments with a shaped reward were scientifically invalid due to data leakage ("reward contamination"). Instead of hiding this, they document it (Appendix A.1) and detail their pivot to a new, valid simulation with a sparse reward. This transparency builds significant trust and strengthens their final conclusions.

2. Clear and Interpretable Problem Formulation: The decision to simplify the action space to a binary choice ('Precision' vs. 'Discovery')  is highly effective. While an abstraction, it perfectly isolates the high-level strategic trade-off at the heart of the cold-start problem. This simplification is what enables the "smoking gun" behavioral analysis (Figure 3)  that reveals the core insight of the paper.

3. Insightful Analysis: The paper does not stop at the "disappointing" result (DQN = Heuristic). It digs deeper to find the reason, using the behavioral analysis to uncover the divergent policies. The resulting "performance ceiling" thesis is a valuable and non-obvious insight for the OPE and RL-for-RecSys communities.

4. Strong Baselines: The comparison is not just against a strawman. The heuristic baseline is the incumbent, production-style rule from prior work, and the authors also include standard baselines like Popularity and Item-kNN. The DQN agent was also rigorously tuned via a grid search to ensure it was a strong competitor.

**Weaknesses:**

1. Simplified Action Space: The paper's main strength is also its primary weakness, which the authors acknowledge. In a real-world system, an agent would not choose either precision or discovery; The binary-choice setup cannot explore this more realistic and nuanced action space, more importantly, we don't know if such discovered insights still make the rules in the more diverse action space.

2. Simulation fidelity from static offline dataset: The simulation is built from a static, historical dataset (MovieLens). A "hit" is defined only as matching a movie the user actually did positively rate in the test set. This design inherently and completely excludes the possibility of rewarding new discoveries, making the "performance ceiling" an almost unavoidable artifact of the experimental design itself.

**Questions:**

Several simple but important questions:
1. The author uses tuned DQN as main operating model for the analysis. However, since DQN is operating on the offline simulation and suffers from low return metric values because of its exploratory behaviors which are actually great, why not try some offline learning methods instead of directly jumping into improving the benchmark?

2. Since the authors claim richer offline simulation or mix of online-offline simulaiton helps on avoiding such incorrect masking between reported metric performance and actural behaviors among baselines, why not compare some simulations (for example, those traditional old offline simulation) to their newly proposed richer simulation or online mixed ones to more soundly demonstrate their conclusions?

3. Authors should compare with more exploratory, advanced, tuned RL baselines such as SAC, PPO, DPO to demonstrarte their claims. Since the time limit, I understand it's not reasonable to compare everything deeply in one paper, just for future suggestions.

---

### Official Review · Reviewer_iwkB · 2025-11-01

**Soundness:** 1
**Presentation:** 1
**Contribution:** 1
**Rating:** 0
**Confidence:** 4

**Summary:**

This work is about reinforcement learning environment for recommender systems.
It is a purely experimental study that compares the performance of DQN with simpler baselines.
The authors present this work as a cautionary tale by providing interpretation on the experimental results.

**Strengths:**

The question of how to test recsys method and how to extrapolate simulations results to live outcomes is a very important topic.

**Weaknesses:**

The paper is not in a shape that would fit a conference like ICLR:
* 6 elements in the bibliography, for a paper at the intersection of two enormous fields (RL, recsys)
* no modeling, no theoretical contribution
* experiments are done on a small scope
-> need to test on more environments, more baselines, more variant
* the claim itself is too vague and not sufficiently backed
* the tone is not adapted for the audience, the beginning of the paper sounds more like a lab journal
* the content itself lack clarity

**Questions:**

see weakness

---

### Meta-Review · Area_Chair_Vv5T · 2025-12-22

**Summary:**

The paper introduces a transparent benchmark for cold-start recommendation and trains a highly optimized DQN in a high-fidelity offline simulator, but finds performance statistically tied with a simple static heuristic on metrics such as cumulative reward and NDCG@10. Despite comparable outcomes, the RL agent learns a substantially more exploratory policy, suggesting that offline simulation can impose a “performance ceiling” that obscures meaningful policy differences and motivating richer offline environments or hybrid online-offline evaluation.

Reviewers identified major flaws in both the methodology and experimental evaluation. Since the authors did not submit a rebuttal, these concerns remain unaddressed.

**Reviewer Concerns:**

The authors did not submit a rebuttal, so all the concerns remain unaddressed.

**Reviewer Scores:**

Since no rebuttal was provided, most reviewers are likely to maintain their original scores. The one more positive review may also be revised downward after the reviewer considers the other reviewers’ concerns.

---

### Decision · Program_Chairs · 2026-01-26

Reject